# Target Lines for *in Planta* Gene Stacking in *Japonica* Rice

**DOI:** 10.3390/ijms23169385

**Published:** 2022-08-20

**Authors:** Ruyu Li, Zhiguo Han, Qian Yin, Meiru Li, Mingyong Zhang, Zhenzhen Li, Ping Wang, Li Jiang, David W. Ow

**Affiliations:** 1Plant Gene Engineering Center, Chinese Academy of Sciences, Guangzhou 510650, China; 2Key Laboratory of South China Agricultural Plant Molecular Analysis and Genetic Improvement, Provincial Key Laboratory of Applied Botany, South China Botanical Garden, Chinese Academy of Sciences, Guangzhou 510650, China; 3University of Chinese Academy of Sciences, 19 Yuquan Road, Beijing 100049, China

**Keywords:** transgenic, Bxb1, integrase, recombinase, GMO

## Abstract

The clustering of transgenes at a chromosome location minimizes the number of segregating loci that needs to be introgressed to field cultivars. Transgenes could be efficiently stacked through site-specific recombination and a recombinase-mediated in planta gene stacking process was described previously in tobacco based on the Mycobacteriophage Bxb1 site-specific integration system. Since this process requires a recombination site in the genome, this work describes the generation of target sites in the *Japonica* rice genome. *Agrobacterium*-mediated gene transfer yielded ~4000 random-insertion lines. Seven lines met the criteria of being single copy, not close to a centromere, not inserted within or close to a known gene or repetitive DNA, having precise recombination site sequences on both ends, and able to express the reporter gene. Each target line tested was able to accept the site-specific integration of a new *gfp*-containing plasmid and in three of those lines, we regenerated fertile plants. These target lines could be used as foundation lines for stacking new traits into *Japonica* rice.

## 1. Introduction

A transgenic trait is typically engineered into a laboratory line that is easy to transform before it is introgressed into elite cultivars. A breeding line for commercial seed production must be homozygous for not only the transgene, but also for the non-transgenic traits associated with each elite cultivar. For diploid and diploid-like allopolyploid plants, (¼)^n^ is the probability for assorting the ‘n’ number of independent linkage units into a homozygous breeding line, provided that there is no linkage drag [1]. Increasing the number of segregating loci would extend the time and labor for each transgene introgression process, especially if the transgene is to be introgressed into a large number of region-specific cultivars.

To maintain a single transgenic locus, developers can combine new transgenes with previously introduced transgenes in vitro for a new round of transformation [2,3,4]. Making a longer transformation construct may not be technically difficult, but to find all the genes appropriately expressed in a single integration event could be challenging. Moreover, it could trigger the need to go through the de-regulation process again for previously introduced traits, since they would be considered a new integration event. 

Bypassing the need for transgene introgression is possible through the direct transformation of elite cultivars. However, most commercial cultivars are difficult to transform, requiring greater effort to obtain a sufficient number of independent transformants for field evaluation. There are also too many locally adapted cultivars that require efficient transformation protocols. Most troublesome from a regulatory perspective is that each commercial cultivar derived from individual transformation of the same DNA would be construed as an independent event requiring individual de-regulation, unlike the deregulation of a single integration event that is then bred out to other field cultivars.

A third approach to keeping transgenes clustered is to insert the new DNA next to previously placed transgenes. This in planta gene stacking can be done, for example, by induction of host-mediated homologous recombination via site-specific nucleases, such as zinc finger nucleases, TALEN (transcription activator-like effector nucleases), meganucleases, and CRISPR (clustered regularly interspaced short palindromic repeats)/Cas proteins that can produce specific genomic DNA breaks to allow for homology directed repair (HDR) from a donor DNA fragment [5,6,7,8,9]. In maize, the herbicide resistance gene *aad1* was stacked next to a pre-existing herbicide resistance gene *PAT* by zinc finger nuclease-induced recombination, with frequencies up to 5% [10]. In cotton, two herbicide resistance genes, *epsps* and *hppd,* were stacked next to pre-existing transgenes *cry2Ae* and *bar* at up to 2% frequency by meganuclease-mediated targeting [11]; and in soybean, four marker genes were stacked into the *FAD2-1a* locus by zinc finger nuclease-mediated non-homologous end joining, yielding three targeted events out of 1290 hygromycin-resistant shoots from immature embryos [12].

Aside from host-mediated homologous recombination, site-specific recombinases can direct the integration of new DNA [13,14,15,16,17,18,19,20,21]. We use a recombinase-mediated in planta gene stacking method that employs the Mycobacteriophage Bxb1 site-specific integration system for integrating new DNA [1,22]. This system consists of the 500 amino acid Bxb1 integrase (recombinase) that catalyzes recombination between an *attP* (phage attachment site, minimal 39 bp) and an *attB* (bacterial attachment site, minimal 34 bp) to generate *attL* (attachment site left) and *attR* (attachment site right) without other proteins or high-energy cofactors [23]. A target site is first created in the plant genome, such as an *attP* site (Figure 1A and Figure 2A). New DNA is introduced through a donor construct that carries two complementary *attB* sequences (Figure 2A,B). Recombination between *attP* with *attB* places the incoming DNA precisely into the genomic target, and two configurations are possible depending on which *attB* site recombines (Figure 2C,D). The preferred configuration can be screened by PCR, and the *attB* not used in the first round of integration can serve as a target site for the next round of integration by a donor plasmid with two *attP* sites. In principle, serial gene stacking can be undertaken by alternating between the uses of *attB* and *attP* donor plasmids (not shown, see [1]). DNA no longer needed after site-specific integration can be removed by the Coliphage P1 Cre-*lox* recombination system, in which the 343 aa Cre (control of recombination) protein recombines directly oriented 34 bp *lox* (locus of x-over) sites. 

Due to the requirement that the plant genome must have a target site for integration of an incoming molecule, one possibility would be to engineer it into the plant genome using site-specific nucleases. Although site-specific nucleases could be used to direct the insertion of the target site into the rice genome, an important consideration was that the target lines should not be restricted from commercial use due to the site-specific nuclease patents. Hence, we screened ~4000 *Agrobacterium*-mediated *Japonica* rice transformants. Seven lines met the criteria of being single copy, not close to a centromere, not inserted within or close to a known gene or repetitive DNA, having precise recombination site sequences on both ends and expressing the reporter gene *gus* (encoding beta-glucuronidase, GUSPLUS version). Each target line was shown capable of accepting the integration of a new plasmid carrying *gfp* (encoding green fluorescence protein, enhanced version) and in three of those lines, regenerated fertile plants were analyzed. Surprisingly, however, not only was *gfp* expression found, but *gus* expression in all three integrant lines was elevated. Since the same newly introduced DNA can reproducibly enhance expression of an adjacent gene in three different chromosome locations, it likely carried an enhancer element.

## 2. Results

### 2.1. Target Constructs

The target construct placed into the rice genome comprised a DNA fragment that encodes *hpt* (encoding hygromycin phosphotransferase) as a selectable marker for transformation, and *gus* as a reporter of gene expression (Figure 1A,B and Appendix A). This DNA fragment was flanked by a set of *lox* sites situated in direct orientation to permit the subsequent removal of the *hpt-gus* DNA after scoring integration events with stable reporter gene expression. At one end, and outside, of the *lox-hpt-gus-lox* fragment was an *attP* site from the Bxb1-*att* site-specific recombination system. The *attP* site permits site-specific recombination with an incoming circular DNA containing an *attB* site from the same site-specific recombination system. On the other side of this *attP* site was a third *lox* site situated in the opposite orientation with respect to the *lox* sites that flank the *hpt-gus* DNA. This third *lox* site permits subsequent Cre-*lox* site-specific excision of unneeded DNA introduced by the incoming molecule that inserts into the target *attP* site. *MRS* recombination sites (pZH37, Figure 1A), or *RS2* recombination sites (pZH36, Figure 1B) flanked the entire DNA segment. The 133 bp *MRS* is recognized by the ParA recombinase from the broad host range plasmid RP4 [24], and the 119 bp *RS2* is recognized by the CinH recombinase from *Acetinetobacter* plasmids [25]. These sites were incorporated into the construct to permit future optional excision of the transgenic DNA internal to these sites, although this study did not test this feature.

### 2.2. Generating Target Sites in the Rice Genome

Hygromycin-resistant plants were regenerated from *Agrobacterium*-mediated transformation of pZH37 and pZH36 (Appendix A), and 1570 out of 2236 pZH37-derived and 1566 of 1717 pZH36-derived plants were *gus* positive by PCR. Two rounds of quantitative real-time PCR (qPCR) were performed to weed out those with high transgene copies. For the 1st round of qPCR on *gus*, 1444 lines showed approximately four or less copies of *gus* DNA. For the second round qPCR of *hpt*, 471 plants showed approximately two or less copies. After subjecting these 471 lines to Southern blotting of SacI cleaved DNA, 188 lines showed a single *hpt* hybridizing band greater than the expected minimum size of 1.9 kb (Figure 1C). The *gus* probe was then used on these 188 clones to reveal that 76 lines showed a single hybridizing fragment greater than the expected minimum size of 6.7 kb (Figure 1D). 

### 2.3. Map Locations

To map the insertion sites, TAIL-PCR was conducted on the 76 single copy lines, of which 62 lines were amplified successfully. A BLAST search against the rice genome database showed that 34 of the 62 were inserted into known genes. Among the other 28 lines, 16 have a precise structure extending through the outermost *lox* recombination sites. The DNA interior to the *lox* sites was not sequenced as it would be deleted after serving its function for identifying clones with stable transgene expression. Of the 16 precise single copy lines, four were discarded because they were located close to a centromere that would be difficult to introgress into other cultivars. Another five insertions were discarded for being close to a gene open reading frame such that the close proximity might affect its expression. This left only seven lines that met the criteria of being a desirable target line. Based on the genome sequence, line TS367 was expected to hybridize to a 3.3 kb band, but a second band was also found. This could be due to partial cleavage by SacI to reveal a 3.76 kb band (from another SacI site 0.46 kb away), or the presence of another (partial) T-DNA left border in the genome (Figure 1C). The lack of another T-DNA was confirmed by Southern analysis of EcoRI-treated DNA. A single left end fragment of >2.4 kb (Figure 1E) as well as a single T-DNA right end band of >6.2 kb was found in all seven lines (Figure 1F). Of these seven lines, four (TS131, TS284, TS325, TS537) were derived from pZH37 (flanked by MRS) and three (TS281, TS367, TS766) were from pZH36 (flanked by *RS2*). 

Figure 1G shows the insertion locations based on matching the left and right border flanking DNA with the rice genome database, available online: http://rapdb.dna.affrc.go.jp/ (accessed on 11 May 2015 and confirmed on 26 August 2021) [26]; Rice-Map [27]. Descriptions at the DNA sequence level are provided in Appendix A. Salient features of each target site are listed below. All insertions are described from the T-DNA left border end to the T-DNA right border end as depicted in Figure 1A,B.

Target site 131 (TS131): Short arm of chromosome II between positions 5,187,244 and 5,187,205. The DNA from 5,187,243 to 5,187,206 is missing, hence line TS131 has a 38 bp deletion at the site of the insertion (Figure 1A and Appendix A). On the left end (same end of T-DNA LB), it lacks the entire 25 bp T-DNA LB plus 26 bp of adjacent vector sequence. On the right end, it lacks 22 bp of the 25 bp T-DNA RB sequence, which is expected after cleavage between bases 3 and 4 of the T-DNA border (Appendix A). Nearest coding regions are 2.9 kb (start codon) and 1.9 kb (stop codon) away from LB and RB, respectively (Appendix A). 

TS284: Long arm of chromosome V between positions 27,877,812 and 27,877,843. The DNA from 27,877,813 to 27,877,842 is not found, representing a 30 bp deletion (Figure 1A and Figure Appendix A). On the left end, it lacks the 25 bp T-DNA LB plus 10 bp of adjacent vector sequence. As expected on the right end, it lacks 22 bp of the T-DNA RB sequence (Appendix A). Nearest coding regions are 0.7 kb (start codon) and 0.9 kb (start codon) away from LB and RB, respectively (Appendix A). 

TS325: Long arm of chromosome I between positions 32,100,641 and 32,100,689. TS325 has a 47 bp deletion at the site of the insertion as the DNA from 32,100,642 to 32,100,688 is not found (Figure 1A and Appendix A). On the left end, the 25 bp T-DNA LB plus 10 bp of adjacent vector sequence are replaced by a 6 bp insertion. On the right end, as expected, it has 3 bp of the T-DNA RB sequence (Appendix A). Nearest coding regions are 2.1 kb (stop codon) and 6.6 kb (stop codon) away from LB and RB, respectively (Appendix A). 

TS537: Long arm of chromosome I between positions 35,913,966 and 35,913,934. Position 35,913,966 is a G nucleotide assigned as host DNA although it is also the same nucleotide if it were assigned as part of the T-DNA left border. There is a 31 bp deletion as the DNA from 35,913,965 to 35,913,935 is missing (Figure 1A and Appendix A). The left end lacks 17 bp of the 25 bp T-DNA LB, and the right end as expected lacks 22 bp of the T-DNA RB sequence (Appendix A). Nearest coding regions are 0.8 kb (stop codon) and 2.8 kb (stop codon) away from LB and RB, respectively (Appendix A). 

TS281: Long arm of chromosome VIII between positions 16,669,159 and 16,669,154. The DNA from 16,669,158 to 16,669,155 is not found representing a 4 bp deletion (Figure 1B and Appendix A). On the left end, 8 bp of the T-DNA LB are replaced with a 10 bp insertion. On the right end, the expected 22 bp missing from the T-DNA RB are replaced by a 24 bp insertion (Appendix A). Nearest coding regions are 4.5 kb (stop codon) and 16.2 kb (start codon) away from LB and RB, respectively (Appendix A). 

TS367: Long arm of chromosome V between positions 27,601,562 and 27,601,606. A 43 bp stretch of DNA from 27,601,563 to 27,601,605 is not found. Host and *RS2* DNA on the left end share 3 bp of identical CGT sequence which we arbitrarily counted as *RS2* DNA (Figure 1B and Appendix A). On the left end, it lacks the T-DNA LB plus 64 bp of adjacent vector sequence. As expected on the right end, it only has 3 bp of the T-DNA RB sequence (Appendix A). Nearest coding regions are 3.0 kb (start codon) and 0.8 kb (start codon) away from LB and RB, respectively (Appendix A). 

TS766: Short arm of chromosome I between positions 9,639,408 and 9,639,426. A 17 bp deletion is found as the DNA from 9,639,409 to 9,639,425 is missing (Figure 1B and Appendix A). On the left end, it lacks 20 bp of the T-DNA LB. As expected, the right end lacks 22 bp of the T-DNA RB (Appendix A). Nearest coding regions are 2.2 kb (start codon) and 3.0 kb (stop codon) away from LB and RB, respectively (Appendix A). 

### 2.4. Bxb1-Mediated Site-Specific Integration into Rice Target Lines

Embryogenic calluses were induced from each of the seven target lines derived from T1 or T2 seeds. The integrating plasmid pZH210B along with pC35S-BNK that produces Bxb1 integrase (Figure 2B) were co-bombarded into the calluses. Site-specific integration of a circular pZH210B molecule into the genomic *attP* site is expected to produce the structure shown in Figure 2C or Figure 2D depending on which *attB* on pZH210B recombines with the genomic *attP*. Among the seven target lines, the putative transformation efficiency, defined by bialaphos selection and GFP activity, ranged from 6% to 19% (Table 1). Pooling the data from all seven target lines, 280 of 2115 calluses were putatively transformed, or a 13.2% putative transformation efficiency comprising those that integrated site-specifically, randomly, a combination of both, as well as those in which bialaphos tolerance and *gfp* expression might have been due to non-integrating molecules. 

The *gfp* expressing calluses were tested by PCR with primers a + b and c + d to detect junctions formed from type I integration, the recombination between the genomic *attP* with the *bar* distal *attB* of pZH210B (Figure 2C), or primers a + e and f + d to detect junctions formed from type II integration with the *bar* proximal *attB* (Figure 2D). From the 280 calluses, 49 were found to harbor at least one set of integration junctions from recombination by either one or both of the *attB* sites in pZH210B (Table 1). Among the 49 PCR positive clones, 32 showed a type I-only integration pattern and this was found from each of the seven target lines. Unexpectedly, only six calluses from three target lines showed a type II-only pattern of recombination. Another 11 clones, spread among four target lines, showed recombination with both *attB* sites, which could indicate different single integration events in different cells within the same callus or different integration events in different homologous chromosomes within a single clone. For these clones with both type I and II integration, the calluses might yield plants harboring only the desired type I structure or might require additional generations to segregate away the undesired type II configuration. Considering that 32 out of 280 GFP positive calluses showed only the preferred type-I-only integration, this 11.4% is still a practical efficiency for obtaining the desired integration event, although this frequency is based on calluses that have been selected by bialaphos and GFP. For unselected calluses, the recovery of 32 preferred type-I-only integration from a total of 2115 bombarded calluses is but 1.5%. 

Attempts were made to regenerate plants from calluses from those target lines that showed type-I-only integration. However, plant regeneration was successful for only TS131, TS325, and TS537, all were from pZH37. Plants did not regenerate from one pZH37-derived line, TS284, and the three pZH36-derived lines; and it is possible that the long bialaphos selection protocol we used might have decreased the regeneration ability of these bombarded calluses. 

### 2.5. Southern Blot Analysis

From TS131, 193 plants regenerated and all but 2 yielded the a + b and c + d PCR products (Table 2). Since they were derived from four independently transformed calluses (A, B, C and D), there has to be at least four independent integration events. Genomic DNA of five T0 plants from each of calluses A, B, C, and D was examined by the *gfp* probe on SacI-treated DNA. As depicted in Figure 2C, the expected internal fragment fusing *gfp* to *gus* should be 9.4 kb, and all 20 plants showed this band (representative callus A plant shown in Figure 2E). 

From TS537, 216 plants regenerated but all were derived from calluses split from a single transformed callus (Table 2), and 204 plants gave the correct PCR recombination junction bands. All five plants chosen for Southern analysis showed a single 9.4 kb SacI band hybridizing to the *gfp* probe (representative plant shown in Figure 2E).

From TS325, 262 plants regenerated from 11 transformed calluses (Table 2), but the correct a + b and c + d PCR junctions were found in only 77 plants from eight calluses. From one to five plants derived from each of six calluses were analyzed by Southern hybridization using the *gfp* probe on SacI-cleaved DNA. A single 9.4 kb band was found from plants derived from two calluses, G and K (representative callus G plant shown in Figure 2E). Plants from the other calluses, B, C, E, and H each showed multiple bands, as though additional copies of pZH210B integrated into the same genome. 

A detailed Southern blot analysis was conducted on a representative T0 integrant plant derived from each target site, TS131-I from callus A, TS325-I from callus G, and TS537-I from callus A. Four DNA probes corresponding to *gfp, bar, hpt*, and *gus* were used against SacI- or XbaI-cleaved DNA (Figure 2C). As expected, none of these probes hybridized to WT (wild type non-transgenic) rice DNA (Figure 2E–H). The *gfp* and *bar* probes should hybridize only to integrant line DNA, whereas *hpt* and *gus* would hybridize to both integrant and target line DNA. 

The *gfp* probe detected the internal 9.4 kb SacI fragment as well as the internal 0.9 kb XbaI fragment in all integrant plants (Figure 2E). The *bar* probe should detect a right end SacI fragment of >4.8 kb and a right end XbaI fragment of >5.2 kb. As the nearest SacI and XbaI sites outside of each target construct are known, the >4.8 kb SacI bands would be 8.0, 9.3, and 11.9 kb, and the >5.2 kb XbaI bands would be 6.5, 10.9, and 8.4 kb, for TS131-I, TS537-I, and TS325-I, respectively (Figure 2C). As shown in Figure 2F, the band sizes detected agreed with expectations. 

The *hpt* probe should hybridize to the same left end band of >1.9 kb in target or integrant DNA when treated with SacI, and same size bands were detected in target and integrant line lanes, as predicted from chromosome sequences of 9.8 kb for 131 and 131-I, 9.2 kb for 537 and 537-I, and 7.9 kb for 325 and 325-I (Figure 2C,G). For DNA cleaved with XbaI, *hpt* should hybridize to a left end fragment of >1.4 kb in both parental and integrant DNA. As shown in Figure 2G, the left end fragments in 131 and 131-I were 7.7 kb, those of 537 and 537-I were 2.9 kb, and those of 325 and 325-I were 3.8 kb, as expected from chromosome sequence calculations. 

The *gus* probe is expected to hybridize to a right end >6.7 kb SacI fragment and a >3.1 kb XbaI fragment (Figure 2A). For lines TS131, TS537, and TS325, based on nearest chromosome sites, respectively, the expected >6.7 kb SacI fragments were 9.9, 11.2, and 13.8 kb, and the expected >3.1 kb XbaI fragments were 4.4, 8.8, and 6.3 kb (Figure 2H). For the integrant plants, the expected hybridization would be a 9.4 kb internal SacI fragment and a 4.1 kb internal XbaI fragment, and these were detected in TS131-I, TS537-I, and TS325-I plants. However, for TS325-I, the same size SacI and XbaI fragments were also detected in the TS325 lane. This would suggest that TS325-I is hemizygous for the type I integration event. As there were no additional bands other than those expected by any of the probes, this shows that there is only a single copy of the transformed DNA at the target site for these integrant lines. We also tested whether the Bxb1 integrase expressing plasmid might have co-integrated into these integrant plants, but primers specific for the Bxb1 integrase gene did not detect a PCR product from TS131-I, TS537-I, and TS325-I genomic DNA, indicating that the Bxb1 integrase gene was not present in these plants (Appendix A)

DNA sequencing of the PCR products from these three integrant lines, from primers a + b, c + d, and g + h (Figure 2C) were all found to be correct. Along the previous sequence confirmation of the left and right ends of the target lines, it should be possible to perform the subsequent Cre-mediated removal of DNA bound by directly oriented *lox* sites, as well as to stack new DNA into the genomic *attB* site located between *gfp* and *bar*.

### 2.6. Expression of Reporter Genes in T2 Integrant Plants

T2 seedlings of these three precise integrant lines along with their parent target lines were examined for *gus* and *gfp* expression (Figure 3). Among the target lines, TS131 showed the lowest GUS activity while TS325 showed the highest, and none showed above background level for GFP. These differences might not necessarily be biologically significant since the differences were at most only two-fold. For the integrant lines, GFP activity was comparably high in TS131-I and TS537-I, but lower in TS325-I. What was striking was the three- to five-fold difference in GUS activity when comparing between target and integrant lines. Since the integration of the *gfp*-*bar* construct pZH201B correlated with elevated *gus* expression, one possibility might be due to a promoter enhancer element within pZH210B. The ScBV (sugarcase bacilliform badnavirus) promoter was used to transcribe *gfp* and the CaMV (cauliflower mosaic virus) 35S RNA promoter was used for *bar*, and one or both of these might have been able to enhance the rice actin2 promoter driving *gus*. A similar observation was found when the expression of the reporter gene *gfp* was elevated by the integration of another construct into a target site in cotton [28].

## 3. Discussion

The reason for developing in planta gene stacking is to enable commercial developers to add new transgenic DNA at an existing transgenic locus. Otherwise, increasing the number of transgenic loci with each new trait added to different locations would increase the workload for introgressing transgenes from a laboratory line to the numerous local cultivars. Recombinase-mediated in planta gene stacking, however, requires the prior placement of a target recombination site in the genome. Homologous recombination-based gene targeting could have been used to direct a target construct to a designated location, as has been demonstrated [20]. However, we used random placement followed by empirical testing of the expression of a reporter transgene. This required more work but insured that the target lines would not be restricted from commercial use due to patented site-specific nuclease technologies. 

This was our first attempt at generating target lines in a major crop and it was an unexpectedly low frequency (0.18%) of finding seven suitable lines out of 3953 plants from *Agrobacterium*-mediated transformation. In our subsequent screening of target lines of other crops, five soybean target lines were obtained from 368 transgenic plants (1.4%) [29], while three cotton target lines were found among 152 transgenic plants (2%) [28]. A likely reason for this discrepancy might be due to the more efficient gene transfer and regeneration of rice. Efficient gene transfer can produce more multiple copy insertions, and indeed, only 1.9% (76/3953; Appendix A) were truly single copy. Efficient regeneration can also produce more clonal plants from the same callus, leading to screening the same event more than once. Nonetheless, we achieved the goal of finding lines where the target construct was not inserted within or close to a known gene, not close to the centromere and showed good expression of the *gus* reporter gene. Each target line was shown to be capable of site-specific integration, although integrant calluses from only three target lines regenerated plants. Moreover, transmission of those integration events was obtained (T2 seedlings, Figure 3). For integrant calluses that failed to regenerate plants, it might be due to having been in culture for too long. 

In our selection of the target lines, we confirmed that the relevant DNA sequences from the outer flanking *MRS* or *RS2* sites to the inner *lox* sites were correct. Although flanking *MRS* or *RS2* sites are not necessary for gene stacking, they were incorporated into the target site to provide future options, and there could be instances where it might be desirable to delete the entire transgenic locus by CinH or ParA recombinase, as, for example, root-specific excision in root crops.

As none of the seven target sites were inserted into or less than 0.7 kb of a nearest gene coding region (Appendix A), it is likely that they would not disrupt host gene function; although only through experience, including introgressed derivatives in field trials can we be certain of a lack of adverse effects on agronomic traits. That is why having multiple target sites available could be of value. 

To remove transgenic DNA no longer needed after site-specific integration, lines obtained from site-specific integration, as in Figure 2C, could be crossed with a *cre*-expressing line, which would require subsequent generations to segregate out the *cre* gene. However, we tested a split *cre* system in *Arabidopsis* in which the *cre* gene is split into *N-cre* and *C-cre* to produce N-terminal and C-terminal Cre polypeptides that are each inactive, but together can reassemble functional Cre activity [30]. In rice, we tested a trait gene linked to *N-cre* for integration into one plant and linked to *C-cre* for integration into another plant. F1 hybrids from the trait-linked *N-cre* and *C-cre* plants produced Cre activity that excised unnecessary DNA, including *N-cre* and *C-cre* as they were also flanked by *lox* sites [31]. This should reduce the generations needed to cross in and out a separate *cre* locus to reset the target locus for the next round of gene stacking.

Of particular interest are recent data that the transgene cassette flanked by the inverted *lox* sites can translocate to the same location in a homologous chromosome or to a new location in a non-homologous chromosome [32]. This effect would break linkage between the transgene locus from nearby DNA, which could expedite introgression of the transgene locus out to field cultivars. 

Transgenes engineered to have different expression patterns might not always be compatible with one another; and indeed, in this study, we showed that targeting new DNA affected the expression of previously placed DNA. Given that possibility, it may be necessary to cluster transgenes with only similar types of expression patterns. If that is the case, then having multiple target sites available would be prudent, as each of these seven target lines might serve as a foundation line for clustering similarly expressed transgenes.

## 4. Materials and Methods

### 4.1. Molecular Constructs

Standard recombinant DNA methods were used. PCR reactions were conducted using Phusion High-Fidelity DNA Polymerase (NEB, Beijing, China). For target constructs, target sites attP, lox, MRS, and RS2 sites were synthetic DNA placed into target constructs pZH37 and pZH36 (Figure 1A,B and Appendix A). Selection gene hpt was controlled by the rice actin1 promoter [33,34] and the CaMV 35S RNA terminator; while gus (gusplus version) was controlled by the rice actin2 promoter [35] and the rice polyubiquitin1 terminator [36]. 

For integration plasmid pZH210B (Figure 2B), bar was controlled by the CaMV 35S RNA promoter and terminator; gfp (enhanced version) was controlled by the ScBV promoter and the octopine synthase gene (ocs) terminator. In pC35S-BNK, the Bxb1 integrase gene was expressed from the CaMV 35S RNA promoter and nopaline synthase gene terminator.

### 4.2. PCR and qRT PCR Analysis

Genomic DNA of plants was extracted and PCR conducted under standard conditions using 2x Mix (GenStar, China) for *gus* detection. Each 10 μL reaction contained 2 × SYBR Premix *Ex Taq*II (TAKARA, Dalian, China), 10 μM forward and reverse primers, and 1 μL plant DNA. Sucrose phosphate synthase gene (*SPS*) was used as the internal reference [37]. To generate standard curves for *SPS*, *gus,* and *hpt*, plasmid pMD-SPS, pZH36 DNA was serially diluted to final concentrations of 10^9^, 10^8^, 10^7^, 10^6^, and 10^5^ copies/μL. Absolute copy numbers of *SPS*, *gus,* and *hpt* in each sample were calculated using Cp values based on the standard curves. Estimated transgene copy number was obtained based on the value of absolute copy number of *gus* and *hpt* divided by absolute copy number of the *SPS* gene. Primer sequences are listed in Appendix A.

### 4.3. Southern Blot Analysis

Genomic DNA (10 ug) cleaved with SacI was transferred to Amersham Hybond-N^+^ membrane (GE Healthcare, Chicago, IL, USA) by 10 × SSC using Model 785 Vacuum Blotter (Bio-Rad, Hercules, CA, USA). [α-^32^P] dCTP-labeled *hpt*, *gus, gfp,* and *bar* fragments with Amersham Rediprime II Random Prime Labelling System (GE Healthcare, Buckinghamshire, UK) were used as hybridization probes. Hybridization and washing methods were according to Sambrook et al. [38]. After washing, the membranes were exposed to a phosphor screen for 5–12 h and scanned on Typhoon FLA 9500 (IP: 635 nm, PMT: 500 V, Pixel size 200 μm). Probe primer sequences are presented in Appendix A.

### 4.4. Target Site Identification and Mapping 

TAIL-PCR on single copy transgene insertion lines was conducted as described [39] with amplification parameters and oligonucleotides for random primers and RB-specific primers adapted from Liu et al. [40]. PCR products were purified from 1% agarose gel for sequencing (Sangon Biotech, Shanghai, China) using primers in Appendix A. TAIL-PCR product sequences were blast searched against the rice genome database from the Rice Genome Annotation Project, the Michigan State University available online: http://rice.plantbiology.msu.edu (accessed on 11 May 2015 and confirmed on 26 August 2021), National Center for Biotechnology Information available online: http://www.ncbi.nlm.nih.gov (accessed on 11 May 2015 and confirmed on 26 August 2021). Vector sequences from TAIL-PCR products we are aligned with the plasmid sequences using online software Clustal Omega, available online: http://www.ebi.ac.uk/Tools/msa/clustalo/ (accessed on 11 May 2015 and confirmed on 26 August 2021). 

### 4.5. Rice Transformation and Site-Specific Integration

*Agrobacterium*-mediated rice transformation (*Oryza sativa* cv. Zhonghua 11) was conducted according to Li et al. [41]. The step-by-step protocol for Bxb1 recombinase mediated site-specific integration in rice has been described in Li et al. [42]. Embryogenic calluses induced from mature embryos of rice target lines and sub-cultured every four weeks were co-bombarded with integration vector pZH210B and the Bxb1 integrase expressing construct pC35S-BNK. DNA was isolated from about 100 mg young rice leaf tissue or transformed callus ground in liquid nitrogen as described [43]. PCR was conducted under standard conditions using 2 x Taq Master Mix (Microanalysis, Emeryville, CA, USA), and gel-purified PCR products were sent out for sequencing (Sangon Biotech, Shanghai, China). 

### 4.6. Transgene Expression

The 50 mg of leaf tissue was ground in 500 uL buffer (50 mM NaPO_4_, pH 7.0, 10 mM β-Mercaptoethanol, 10 mM Na_2_EDTA, 0.1% sodium lauryl sarcosine, 0.1% Triton). GUS and GFP enzyme activities were assayed as described [44] and normalized to protein concentration determined by a Bradford Protein Assay Kit (Thermo Fisher, Pierce, MO, USA). GUS activity by staining was conducted on tissues covered with X-GluC solution (50 mM sodium phosphate buffer pH 7.0, 10 mM EDTA pH 8.0, 0.1% (*v*/*v*) TritonX-100, 0.5 mg/mL X-GluC) and incubated at 37 °C for 12–16 h. GFP assayed for fluorescence was visualized using a DMI6000B microscope (Leica, Wetzlar, Germany); excitation filter from 440 to 520 nm, and 510 LP barrier filter were used. For quantifying GFP activity, 200 μL of protein extraction was placed into black 96-well plates and detected by Mithras LB 940 Multimode Microplate Reader (Berthold, Bad Wildbad, Germany), with 470 nm excitation wavelength and 509 nm emission wavelength, counting 1.00 s.

## 5. Deposition in GenBank

The construct sequences of the pZH37, pZH36, and pZH210B were deposited in GenBank, with accession numbers OK632017, OK563730, and OK632018, respectively.

## 6. Patents

A China patent was issued (CN 104673824 A).

## Figures and Tables

**Figure 1 ijms-23-09385-f001:**
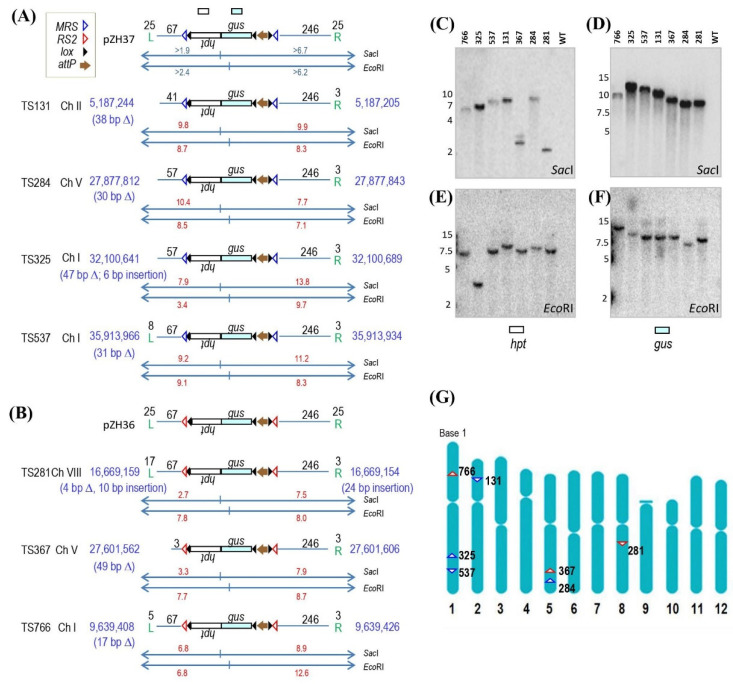
Genome structure and location of rice target lines. (**A**,**B**) Schematic representation of target site structures derived from pZH37 (**A**) and pZH36 (**B**). Blue lettering indicates chromosome positions at left and right ends and, if any, chromosome deletions and insertions in parentheses. Black lettering above L (left) and R (right) T-DNA borders, if present, show the number of L and R bp; between *MRS* or *RS2* to T-DNA ends show number of vector bp. DNA probes *hpt* and *gus* shown above pZH37; restriction map for pZH36 is the same as for pZH37. Target line restriction maps calculated from chromosome restriction sites outside of transgenic DNA and sizes shown in red lettering. (**C**–**F**) Southern blots detected single left end fragment in SacI (**C**) or EcoRI (**E**) cleaved genomic DNA when probed with *hpt* and single right end fragment in SacI (**D**) or EcoRI (**F**) treated DNA when probed with *gus*. WT is *Oryza sativa* cv. Zhonghua 11; size markers on left side of each blot (based on ethidium-bromide-stained gels, not shown). Hybridization probes shown above pZH37; recombination sites indicated in legend; fragments and size markers in kb; genes transcribe left to right except for *hpt* indicated by upside-down lettering; promoters and terminators not shown, see Materials and Methods. (**G**) Rice genome map from www.ricemap.org; genome coordinates start at the top for each chromosome. Target sites derived from pZH37 and pZH36 shown as *MRS* and *RS2* sites, respectively, with site orientation indicating the orientation of insertion.

**Figure 2 ijms-23-09385-f002:**
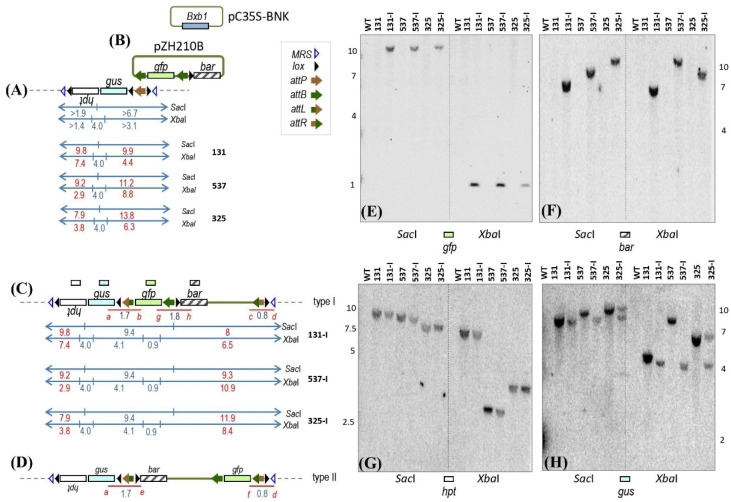
Structure of T1 target and T0 integrant lines. Not-to-scale depiction of recombination between genomic *attP* in rice target line (**A**) and either the *gfp*-upstream *attB* or the *gfp*-downstream *attB* in pZH210B (**B**) to produce the integrant structure in (**C**) or (**D**), respectively. Recombination catalyzed by co-transforming Bxb1 integrase-expressing construct pC35S-BNK. (**E**–**H**) Southern blots of SacI or XbaI cleaved genomic DNA probed with *gfp*, *gus*, *hpt* or *bar* as indicated. WT as in Figure 1; size markers on the side of each blot (based on ethidium-bromide-stained gels, not shown). Predicted border fragment sizes shown in red lettering were calculated from the nearest chromosomal SacI or XbaI; genes as in Figure 1 except *gfp* encoding green fluorescent protein and *bar* for bialaphos resistance; fragment sizes in kb; red lines represent PCR products.

**Figure 3 ijms-23-09385-f003:**
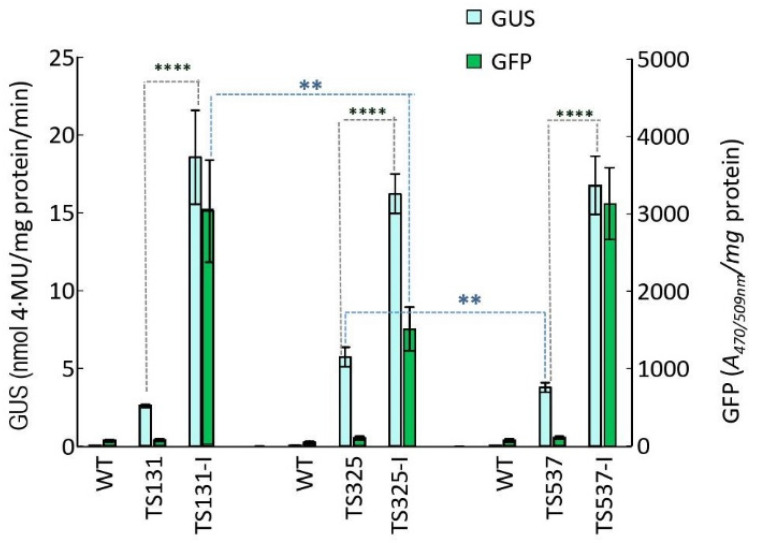
Expression of reporter genes in T2 integrant rice seedlings. T2 homozygous seedlings (TS131-I, TS325-I, TS537-I) were grown in nutrient solution for 21 days before assaying for GUS and GFP activity. Controls were WT (as in Figure 1) and T1 homozygous target line seedlings (TS131, TS325, TS537). Values are mean ± SD *n* = 6 to 9, (**) *p* < 0.01; (****) *p* < 0.0001, Fisher’s Least Significant Difference (LSD).

**Table 1 ijms-23-09385-t001:** Detection of site-specific integration in rice calluses.

Target Lines	Number of Bombarded Calluses	* Number of Transformed Calluses (Transformation Efficiency)	PCR of Recombination Junctions							Site-Specific Integration Efficiency
			Calluses			Type I		Type II		
			Type I Only	Type II Only	Type I + II	a + b	c + d	a + e	f + d	
TS131	30	5 (17%)	4			+	+	−	−	80% (4/5)
TS284	90	7 (8%)	1			+	+	−	−	14% (1/7)
TS325	765	145 (19%)	14			+	+	−	−	16% (23/145)
				2		−	−	+	+	
					7	+	+	+	+	
TS537	330	21 (6%)	2			+	+	−	−	24% (5/21)
				2		−	−	+	+	
					1	+	+	+	+	
TS281	315	29 (9%)	4			+	+	−	−	14% (4/29)
TS367	285	25 (9%)	4			+	+	−	−	20% (5/25)
					1	+	+	+	+	
TS766	300	48 (16%)	3			+	+	−	−	15% (7/48)
				2		−	−	+	+	
					2	+	+	+	+	
Total	2115	280 (13.2%)	32			+	+	−	−	11.4% (32/280)
				6		−	−	+	+	2.15% (6/280)
					11	+	+	+	+	4.0% (11/280)

* bialaphos resistant and GFP-positive calluses over 5 mm in size; recombination junctions from primer sets a + b, c + d; a + e, f + d shown in Figure 2; + (or −) = detected or (not detected).

**Table 2 ijms-23-09385-t002:** PCR and Southern detection of rice plants with type I site-specific integration.

Target Lines	Different Calluses	Number of Integrant Plants Tested by PCR	Junction PCR (a + b)	Junction PCR (c + d)	Number of Integrant Plants Tested by Southern	Number of *gfp* Copy
TS131	A	61	+	+	5	1
	B	34	+	+	5	1
	C	36	+	+	5	1
	D	60	+	+	5	1
		2	−	−	0	nd
Subtotal		193	191	191	20	1
TS537	A	204	+	+	5	1
		2	+	−	0	nd
		10	−	−	0	nd
Subtotal		216	206	204	5	1
TS325	A	6	−	−	0	nd
	B	4	+	+	4	5
		22	−	−	0	nd
	C	2	+	+	2	2 to 3
		38	−	−	0	nd
	D	1	+	+	0	nd
		4	+	−	0	nd
	E	2	+	+	2	2
		1	−	+	0	nd
		27	−	−	0	nd
	F	36	−	+	0	nd
		18	−	−	0	nd
	G	19	+	+	5	1
	H	2	+	+	1	4
		1	−	+	0	nd
		16	−	−	0	nd
	I	2	+	+	0	nd
		1	−	+	0	nd
		8	−	−	0	nd
	J	4	−	−	0	nd
	K	45	+	+	5	1
		1	+	−	0	nd
		2	−	−	0	nd
Subtotal		262	84	116	19	1~5
Total		671			44	1~5

+ (or −) = detected (or not detected) PCR product; nd = not determined.

## Data Availability

All relevant data are included within this article and its Appendix A, except for construct sequences deposited in GenBank. Constructs, bacterial strains, and seed stocks are available from the corresponding author R. Li upon request.

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
