# Peer review of "Target Lines for in Planta Gene Stacking in Japonica Rice"

_ijms, 2022, doi:10.3390/ijms23169385_

Round 1

Reviewer 1 Report

In the manuscript entitled “Target lines for in planta gene stacking in Japonica”, the authors used a recombinase-mediated in planta gene stacking technique via the Mycobacteriophage Bxb1 site-specific integration to introduce foreign DNA into Japonica genome. After Agrobacterium-mediated transformation with pZH37 and pZH36, hygromycin-resistant plants were obtained. Using a combination of PCR, qPCR and Southern blotting 76 plant lines were obtained with a single hybridizing fragment greater than the expected minimum size of 6.7 kb. Of the 76 lines, seven contained a single copy of the transgene. The transgene was not integrated in a gene or repetitive DNA, located distally from a centromere and on both ends were sequences of the recombination sites. Additionally, the seven lines accepted gene stacking with a construct carrying a GFP gene and fertile plants were regenerated from three of these lines.

In planta gene stacking is a beneficial technique for crop developers to integrate additional transgenes into existing transgenic locus. Thus, this study provides seven Japonica rice lines for developers to integrate new desirable rice traits.

The manuscript is well-written, organized and simple to understand. I did not detect any major issues with the manuscript or the design of the study. I only have a few minor suggestions.

·       Consistency – sometimes the authors italicized Japonica and at other times they did not.

·       The manuscript can be improved by showing diagrams of the constructs described in section 2.1 in their entirety.

·       Sentence in lines 389 – 391 is awkward, consider revising the sentence.

Author Response

Thanks for your kind review. We sincerely hope it could be a beneficial technique for crop developers.  We made revisions point-by-point for your suggestions as follows:

  • Consistency – sometimes the authors italicized Japonica and at other times they did not.

         Corrected.

  • The manuscript can be improved by showing diagrams of the constructs described in section 2.1 in their entirety.

         Figure S1 for constructs pZH36 and pZH37 was added in supplementary info.

  1. Sentence in lines 389 – 391 is awkward, consider revising the sentence.

        The sentence was  revised as "Each target line was shown to be capable for site-specific integration, although integrant calluses from only three target lines regenerated plants".

Reviewer 2 Report

The manuscript by Ruyu Li et al does an excellent job demonstrating Agrobacterium-mediated gene transfer which yielded seven recombinant sites in the Japonica rice genome.  This work presents a paradigm that can be broadly and usefully applied. I believe the importance of this paper stems from the applicability of the approach to the several thousand of new genes that Next-Gen sequencing will uncover in the next few years and the challenge we will have in figuring out the function of these genes and their resulting transfer.

Author Response

Thanks for your kind review and great encouragement.  We sincerely hope it could be a beneficial technique for the research community.